# Assessment of Sleep-Related Problems in Children with Cerebral Palsy Using the SNAKE Sleep Questionnaire

**DOI:** 10.3390/children8090772

**Published:** 2021-09-01

**Authors:** Larissa Alice Dreier, Tugba Kapanci, Katharina Lonnemann, Margarete Koch-Hogrebe, Lucia Wiethoff-Ubrig, Markus Rauchenzauner, Markus Blankenburg, Boris Zernikow, Julia Wager, Kevin Rostasy

**Affiliations:** 1Department of Children’s Pain Therapy and Paediatric Palliative Care, Faculty of Health, School of Medicine, Witten/Herdecke University, 58448 Witten, Germany; l.dreier@pedscience.de (L.A.D.); m.blankenburg@klinikum-stuttgart.de (M.B.); b.zernikow@pedscience.de (B.Z.); j.wager@pedscience.de (J.W.); 2PedScience Research Institute, 45711 Datteln, Germany; 3Department of Pediatric Neurology, Children’s and Adolescents’ Hospital Datteln, Witten/Herdecke University, 58448 Witten, Germany; t.kapanci@kinderklinik-datteln.de (T.K.); k.lonnemann@web.de (K.L.); m.koch-hogrebe@kinderklinik-datteln.de (M.K.-H.); l.wiethoff-ubrig@kinderklinik-datteln.de (L.W.-U.); 4Department of Pediatrics, Hospital Kaufbeuren, 87600 Kaufbeuren, Germany; markus.rauchenzauner@kliniken-oal-kf.de; 5Department of Pediatrics, Medical University Innsbruck, 6020 Innsbruck, Austria; 6Paediatric Neurology, Psychosomatics and Pain Therapy, Center for Child, Youth and Women’s Health, Klinikum Stuttgart, Olgahospital/Frauenklinik, 70174 Stuttgart, Germany

**Keywords:** cerebral palsy, sleep, SNAKE, GMFCS, BFMF, children

## Abstract

Cerebral palsy (CP) represents the most common motor impairment in childhood. The presence of sleep problems has not yet been investigated with an instrument specifically designed for this population. In this hospital-based, prospective study, N = 100 children (M = 7.9, range: 2–18 years) with CP were included. All patients underwent pediatric neurologists’ screening incorporating instruments (Data Collection Form; Gross Motor Functions Classification System, GMFCS; Bimanual Fine Motor Function, BFMF) recommended by the “Surveillance of Cerebral Palsy in Europe (SCPE)”. Parents completed the “Sleep Questionnaire for Children with Severe Psychomotor Impairment (SNAKE)”. Children’s sleep behavior was increasingly conspicuous, with greater gross motor (SNAKE scales: disturbances remaining asleep, daytime sleepiness) and fine motor (additionally SNAKE scale arousal and breathing problems) functional impairment. Overall, a proportion of children showed sleep behavior outside the SNAKE’s normal range. No relevant sleep differences were identified between different CP subtypes and comorbidities. Applying a population-specific questionnaire, children’s functional impairment seems to be more relevant to their sleep behavior than the CP subtype or CP comorbidities.

## 1. Introduction

Cerebral palsy (CP) constitutes the most common cause of motor abnormalities in infants and children, describing a group of complex permanent motor and posture disorders that result from lesions, abnormalities, or non-progressive interferences of the developing fetal or immature brain [1,2,3].

Based on children’s neurological signs and the classification of the “Surveillance of Cerebral Palsy in Europe” (SCPE) [1], different subtypes, such as spastic, dyskinetic, dystonic, ataxic, or mixed forms, can be differentiated [1,2]. In relation to a child’s functional limitations, need for assistive technology, and quality of movement, motor function can be classified according to the Gross Motor Function Classification System (GMFCS) and thus be assigned to one of a total of five levels, considering the child’s age [4]. Children’s fine motor skills, on the other hand, can be defined with the help of the likewise five-level Bimanual Fine Motor Function System (BFMFS), which focuses on bimanual functional limitations [5].

CP is often accompanied by a multitude of comorbidities, such as epilepsy, visual, hearing, or cognitive impairment [1,6]. With a prevalence of around 19–63% [1,7,8,9,10,11,12,13,14,15,16], sleep problems constitute a common phenomenon in this population, possibly triggered by the aforementioned CP comorbidities.

Sleep problems are not only having a major impact (e.g., learning or behavioral difficulties) on the well-being of the affected child but are also frustrating for parents and siblings and may lead to a deterioration of their own sleep [6,11,17,18,19].

A simple but efficient way to obtain information regarding children’s sleep is the use of sleep questionnaires. Questionnaire-based surveys have shown that sleep problems appear to be cumulative [11,17,20] and, when considered in detail, may manifest, for instance, in the form of difficulties falling asleep [13,17,21,22,23,24], maintaining sleep [17,21,22,23,24], sleep arousals [13,17,21,22,24], altered sleep-wake patterns [17], or breathing problems [13,25]. Thereby, the number of sleep problems seems to be related to the severity of the CP phenotype and the parallel presence of comorbidities [17].

The “Sleep Questionnaire for Children with Severe Psychomotor Impairment” (Schlaffragebogen für Kinder mit Neurologischen und Anderen Komplexen Erkrankungen, SNAKE) was specifically developed for children with psychomotor impairments and complex (neurological) conditions and was confirmed to show a good reliability and validity [26]. It comprises items that are disregarded in other frequently used questionnaires, such as the “Sleep Disturbance Scale for Children” (SDSC; [21,27]), the “Children’s Sleep Habits Questionnaire” (CSHQ; [28,29,30,31,32]), or the “Sleep Behavior Questionnaire” (SBQ; [33,34]), that have been applied in different studies [35] but, however, were developed per se for healthy persons [26,36,37]. Although the potential utility of the SNAKE for examining sleep in children with CP has been pointed out [38], it has not yet been used in a systematic study with this particular population.

Therefore, the aim of this study was to investigate the sleep and potential sleep problems in children with different CP subtypes considering the potential impact of degree of motor impairment and CP comorbidities.

## 2. Materials and Methods

### 2.1. Participants

For this study, children aged between 2 to 18 years fulfilling the diagnostic criteria for CP according to the “Surveillance of Cerebral Palsy in Europe” (SCPE) [1]) were included. The study was approved by the Ethics Committee of the Witten/Herdecke University, Germany (approval code: 172/2014). All parents gave informed consent to participate in the study.

### 2.2. CP Screening

Comprehensive screening was performed on all children with CP who were planned to be admitted to the Pediatric Neurology ward of the Children’s and Adolescents’ Hospital Datteln in Germany between the years 2015–2017. Reasons for admission included symptom control (e.g., seizures), neuro-orthopedic assessments, cognitive testing, or other issues, such as targeted weight gain.

For this, all children whose parents consented to participate in the study were examined by a board-certified pediatric neurologist (K.R., M.K., L.W.-U.) who then, in the first step, assigned a CP diagnosis based on the SCPE classifications including the following subtypes: bilateral spastic cerebral palsy (BSCP), unilateral spastic cerebral palsy (USCP), dyskinetic cerebral palsy (DCP), or hypotonic-ataxic cerebral palsy [1,39].

Additional patient-related data were systematically recorded by the neuropediatricians using the SCPE “Data Collection Form for Cerebral Palsy” [39]. These data included the children’s gestational age (categorized into extreme preterm delivery: fewer than 32 weeks of gestation; preterm delivery: 32–36 gestational weeks; term born: 37–41 gestational weeks; postterm: more than 41 gestational weeks), birth weight (below 1500 g, 1500–2500 g, more than 2500 g), type of delivery (vaginal, planned C-section, emergency C-section), need for referral to the neonatal intensive care unit (NICU; yes, no), presence of cognitive impairment (yes, no, not known), presence of epilepsy (defined by two unprovoked seizures; yes, no), current intake of anticonvulsants (yes, no, not known), and presence of (severe) hearing and/or (severe) visual impairment (yes, no). Severe visual impairment was defined as a visual loss level <6/60 (Snellen scale, ref. [40]) or <0.1 (decimal scale, ref. [41]) in both eyes (blindness or no useful vision, after correction, in the better eye, ref. [1]). A severe hearing impairment was present with a hearing loss of >70 decibels (before correction, on the better ear; ref. [1]). For the assessment of the children’s cognitive, visual, and hearing abilities, the definitions of the SCPE Data Collection Form for Cerebral Palsy were considered adequate for the study purpose so that more complex assessment instruments, such as the Manual Ability Classification System (MACS, ref. [42]), did not need to be involved.

To classify the patients’ CP severity on a functional level, the GMFCS and the BFMF were applied [4,5,43].

### 2.3. Patients’ Sleep Behavior

To assess patients’ sleep behavior, the “Sleep Questionnaire for Children with Severe Psychomotor Impairment” (SNAKE) was handed out to the parents at admission. The SNAKE assesses children’s sleep behavior during the past 4 weeks. It is based on parent’s or caregiver’s proxy reporting and includes five scales oriented on the International Classification of Sleep Disorders-2 [44]: (1) disturbances going to sleep (value range: 5–20; normal range: 6–14, T-score: 40–60), (2) disturbances remaining asleep (value range: 5–20; normal range: 8–17 points, T-score: 41–60), (3) arousal and breathing disorders (value range: 6–24; normal range: 7–16, T-score: 41–60), (4) daytime sleepiness (value range: 3–12; normal range: 4–9, T-score: 41–58), and (5) daytime behaviour disorders (value range: 4–16; normal range: 4–12, T-score: 36–59; [26,36,37,45]. In addition, a global rating of the child’s sleep during the reference period can be made (4-point Likert scale; very well, well, satisfactory, poor).

### 2.4. Data Analysis

Since the assumption of normal distribution had to be discarded for all variables (Kolmogorov–Smirnov test; all *p* > 0.05), non-parametric statistics were applied for analyses. No a-priori sample size calculation was conducted. The sample size was set to N = 100. Due to missing sample size calculation, sensitivity analyses were performed for each test via the software G*Power, a free-to-use tool for calculating statistical power for a variety of statistical tests [46]. For this, a small to medium effect size and sufficient statistical power of 0.9 was assumed by default.

The patients’ demographic data (age, sex) were extracted from the patient’s medical records and, like the CP screening data, analyzed descriptively. The SNAKE’s mean scale values were determined to evaluate the patients’ sleep behavior.

A Kruskal–Wallis test followed by pairwise comparisons were used to determine potential differences in the sleep patterns among age groups, which were defined as follows: toddler (age 2–3 years), child (age 4–12 years), and adolescent (age 13–18 years).

The global rating of a child’s sleep quality was dichotomized into very well/well and satisfactory/poor. Point-biserial correlations were conducted to investigate the relationship between the five SNAKE scales and the remaining variables. Because of an expected cell frequency less than 5, the Fisher–Freeman–Halton exact test followed by Bonferroni–Holm correction was applied to examine associations between the GMFCS/BFMF, global rating of a child’s sleep quality, and CP comorbidities. Paired comparisons and group differences regarding the presence and extent of sleep disorders detected by the SNAKE (e.g., between the subtypes of cerebral palsy) were investigated via the Kruskal–Wallis test and the Mann–Whitney U test with subsequent Bonferroni–Holm corrections.

All analyses were performed two-tailed, with *p*-values < 0.05 indicating statistical significance using the IBM SPSS statistics software (version 27).

## 3. Results

### 3.1. Patient Characteristics

A total of *n* = 100 children (*n* = 41 female, *n* = 59 male) aged 1 to 18 years (M = 7.9; SD = 4.46; range: 2–18 years) were included in the study. Detailed patient information is depicted in Table 1.

Most of the children (*n* = 51, 51.5%) were born full-term, with birth weight exceeding 2500 g (*n* = 51, 53.1%). The majority were delivered by emergency C-section (*n* = 41, 41%;) and required intensive care in a NICU after birth (*n* = 68, 68%), where *n* = 44 (44%) received ventilation by a respirator or continuous positive airway pressure (CPAP) for more than 24 h.

BSCP represented the most common CP type (*n* = 57, 57%). The leading reason for CP was periventricular leukomalacia (*n* = 25, 25%). The most common GMFCS level was level 5 (*n* = 31, 31%), whereas most children achieved BFMF level 1 (*n* = 35, 35%; Table 1).

Regarding potential CP comorbidities, *n* = 58 (58%) children experienced epilepsy that led to the use of antiepileptic drugs in *n* = 47 (81%) children at the time of data collection. A total of *n* = 23 (23.3%) children experienced hearing and *n* = 65 (65.7%) visual impairment. Varying severity of cognitive impairment was present in *n* = 56 (71.8%) children.

Significant relations were identified between the GMFCS/BFMF and the CP comorbidities severe visual impairment, cognitive impairment, and epilepsy. The BFMF additionally showed a significant association with hearing impairment (Fisher–Freeman–Halton exact test; detectable effect size: 0.4; after Bonferroni–Holm correction, all *p* < 0.05).

### 3.2. Patients’ Sleep Behavior

In 70% (*n* = 70) of the cases, the SNAKE was filled out by a child’s mother (father: *n* = 15, 15%; mother and father: *n* = 12, 12%; other relatives: *n* = 3, 3%). On average, children achieved the following SNAKE scale values: (1) disturbances going to sleep: M = 10.81, SD = 3.83; (2) disturbances remaining asleep: M = 11.11, SD = 4.47; (3) arousal and breathing disorders: M = 10.04, SD = 3.68; (4) daytime sleepiness: M = 5.63, SD = 2.76; and (5) daytime behavior disorders: M = 8.35, SD = 3.49. Skewed distributions or outliers could not be found for any of the scales. On all scales, a certain proportion of children achieved mean SNAKE scale values outside the normal range: (1) disturbances going to sleep: *n* = 24, 24.7%; (2) disturbances remaining asleep: *n* = 35, 35.4%; (3) arousal and breathing disorders: *n* = 32, 32%; (4) daytime sleepiness: *n* = 52, 53.1%; and (5) daytime behavior disorders: *n* = 17, 17.2% (Figure 1).

On scale 4 (daytime sleepiness), a significant difference was identified in the sleep patterns of the participants depending on their age. Pairwise comparisons showed that toddlers achieved significantly higher SNAKE scale 4 values than the comparison groups children (χ^2^ = 28.57, *p* < 0.01) and adolescents (χ^2^ = 30.98, *p* < 0.01; detectable effect size: 0.4).

Overall, *n* = 29 (29.3%) parents/caregivers stated that their child had slept very well during the past 4 weeks, *n* = 49 (49.5%) that their child had slept well, *n* = 17 (17.2%) that their child had slept satisfactorily, and *n* = 4 (4%) that their child had slept poorly. The SNAKE’s global rating of a child’s sleep quality was related to a child’s hearing impairment (Fisher–Freeman–Halton exact test; after Bonferroni–Holm correction, *p* < 0.05).

Significant positive correlations were found between four of the five scales and the extent of gross motor impairment as measured by the GMFCS. Likewise, the degree of fine motor impairment (BFMF) was positively correlated with all five SNAKE scales. Interestingly, no correlations between the type of CP and the SNAKE scales were identified (all *p* > 0.05, Table 2).

Regarding the assessed comorbidities, the majority of parents of children with epilepsy indicated in the SNAKE that their child did not sleep poorly because of epileptic seizures (*n* = 41, 74.5%, SD = 0.91). Further, the SNAKE scale 2 (disturbances remaining asleep) correlated with (severe) visual and cognitive impairment as well as with epilepsy. For scale 3 (arousal and breathing disorders), correlative associations with hearing impairment and epilepsy were identified. Scale 4 (daytime sleepiness) correlated with severe hearing impairment and epilepsy and scale 5 (daytime behavior disorders) with cognitive impairment and epilepsy. None of the SNAKE scales showed an association with anticonvulsant medication intake (Table 2; detectable effect size: 0.3).

Children with the least gross motor impairment achieved significantly lower scale values on scale 2 (disturbances remaining asleep) than those with more profound motor deficits (χ^2^ = 26.79, *p* < 0.01; pairwise comparisons: GMFCS 1–3, GMFCS 1–4, GMFCS 1–5; all *p* < 0.05), as can be seen in Figure 2 (detectable effect size: 0.4). A similar result could be demonstrated for scale 4 (χ^2^ = 17.39, *p* < 0.05; pairwise comparisons: GMFCS 1–5, *p* < 0.001; GMFCS 3–5, *p* < 0.05; GMFCS 4–5, *p* < 0.05; Figure 2a).

Pairwise comparisons as per the post-hoc test of the Kruskal–Wallis test showed that children in the greatest fine motor impairment group (BFMF 5) achieved higher scores on the scales 2 (χ^2^ = 32.13, *p* < 0.01), 3 (χ^2^ = 14.09, *p* < 0.01), and 4 (χ^2^ = 14.80, *p* < 0.01) of the SNAKE questionnaire than children in the other groups (Figure 2b).

After Bonferroni correction, the conducted Mann–Whitney U tests (detectable effect size: 0.6) showed no differences between children experiencing severe visual impairment, reported cognitive impairment, or epilepsy and those children who did not.

## 4. Discussion

This study aimed to investigate the sleep of children with different types and comorbidities of CP using a sleep questionnaire specifically designed for those with neurological and other complex conditions. Similar to already existing evidence, we identified an association between GMFCS/BFMF and different CP comorbidities, which again illustrates the complex interplay between functional abilities and the concomitant symptoms of the CP disease. A plausible and prominent explanation for this may be that different brain lesions causing motor dysfunction may also have direct and indirect effects on other functional areas, such as speech, hearing, or epilepsy [20,47,48,49,50,51].

One of the study’s main findings was that apparently not the CP disorder per se or the corresponding CP subtype is crucial for sleep or potential sleep problems in this particular population but much more the child’s functional impairment as measured by the GMFCS or BFMF. A similar finding regarding the association between the gross motor impairment and sleep behavior of children with CP was described in a recently published study [15].

Unlike in our study, the authors found an association between the “disorders of arousal” scale of the “Sleep Disturbance Scale for Children” (SDSC, ref. [27]) and CP children’s gross motor impairment. One possible explanation for this difference may be that none of the existing studies so far used an instrument for this very special population, a consideration also made by the authors of the study mentioned above [15].

(Sleep) questionnaires that have not been developed and validated for children with exceptional motor characteristics always bear the risk that (motor) behaviors that are somewhat normal for the child and are possibly not considered by parents to be troublesome for the child’s sleep might nevertheless be rated as sleep problems in such questionnaires. This consideration also fits with our finding that the children included in this study were within the normal range on most of the SNAKE scales. This is confirmed by only 4% of parents who rated their child’s sleep in the last 4 weeks as satisfactory or poor overall. In contrast to other questionnaires currently in use, it is conceivable that the SNAKE is capable of appropriately contextualizing the sleep behavior of children with underlying CP disorders and does not “false positively” declare it a disorder. This strongly suggests that for special populations, particular instruments should always be used [15,52]. We identified an association between the children’s global sleep quality and the presence of a hearing impairment. Today, it is known that maternal voice can positively influence children’s sleep and, for example, prevent involuntary awakening due to environmental influences [53]. Due to a hearing impairment, parents may not or only limitedly be able to acoustically soothe their child or to perform popular sleep rituals, such as reading a book aloud. This may be reflected in the parental assessment of their child’s sleep quality.

Our study showed that toddlers showed a more conspicuous daytime sleepiness than children or adolescents. This contradicts existing literature, which identified more sleep problems in older children than in younger ones [54]. Since none of the other SNAKE scales showed corresponding effects, it is unlikely that this result can be explained by, for example, toddlers’ peculiar sleep-regulation mechanisms. Possibly, parents misinterpreted the questions about their child’s sleep behavior during the daytime and considered “intentional” falling asleep like the nap, which is typical for this age group, as sudden (pathological) falling asleep during the daytime. Future projects should shed more light on such findings by collecting additional sleep-associated variables (e.g., does the child take naps, does the child take medication during the day).

We found that children with greater gross and fine motor impairment showed more sleep-through difficulties and more frequent daytime sleepiness than children with lesser impairments, which, focusing on the specific type of sleep problems, fits the existing literature [25,54]. At the same time, this seems plausible for children with CP, whose nightly sleep may be disturbed, e.g., by being physically unable to independently position their bodies and thus becoming uncomfortable, by nightly pain, or by medical or nursing measures, and consequently may lead to associated daytime sleepiness [6,20,23,24,55,56].

Although we identified associations between children’s sleep patterns and CP comorbidities, post-hoc analyses showed no relevant differences in the sleep behavior of children with and without these comorbidities. Probably, it is not so much the CP comorbidities but the children’s functional impairments that are relevant to their sleep. Nevertheless, since other studies have already shown comparable associations, one possible reason for this may simply be that our sample size was still too small. Future studies involving more detailed information on comorbidities (e.g., by using objective measures, such as actigraphy, or by qualitative interviews with parents and providers regarding their assessment of the importance of these factors for children’s sleep) and a larger sample size should be conducted [3,16,23,54].

## 5. Limitations

In our study, some children also scored outside the normal range of the SNAKE, which may tend to indicate abnormal sleep behavior and fits with the numerous existing studies, which ascribed sleep disturbances to children with CP [17,21,25,54,57,58]. Nevertheless, as the greatest limitation of the study, it must be stressed at this point that the SNAKE defines no cut-off values that determine at what point a child’s sleep behavior is clinically relevant and thus potentially requiring special attention or treatment [37,45]. In order for the SNAKE to be fully applicable to complex samples, such as the one studied, and also to have relevance as a diagnostic tool for everyday clinical practice, the definition of cut-off values, for example, as part of a revision of the SNAKE, should definitely be sought.

Another limitation is that the study’s sample size (N = 100) was not based on a-priori calculations. Even though the sensitivity calculations showed that sufficient effect sizes and thus interpretable and meaningful results could be expected for all tests of interest, corresponding a-priori calculations should be performed in future replicating studies. In this context, it is nevertheless important considering that especially in highly specialized settings, such as the one focused on in this study, researchers are always faced with the challenges of small samples [59]. Researchers should therefore always weigh realistically between the desired sample sizes and effects and the actual sample that is presumably to be reached.

The sleep behavior of children with CP may be influenced by more parameters and patient characteristics that were not evaluated in detail within the scope of this study. For example, information on the (artificial) respiration of a child, on the child’s entire medication, on cardiorespiratory, or musculoskeletal abnormalities were not focused in detail, as we were primarily interested in variables that are recorded as part of the SCPE Data Collection Form and by the SNAKE. For the purpose of the study, the selected variables are to be considered satisfactory, as the main objective of the study and our corresponding questions could be answered. Nonetheless, CP is a highly complex disorder [3], so the collection of additional patient characteristics and variables is imperative in future studies to continuously better understand sleep behavior and potential sleep problems in this population [6,57,58].

Lastly, the sleep-related results of our study are based on the judgment of the children’s parents. Given that these are experts on their child and that the application of sleep questionnaires, albeit not specifically developed for children with neurological disorders and psychomotor impairments, is a frequently chosen method in children and patients with CP [54], this should not per se be considered a weakness of the study. Even if there are more objective procedures, such as polysomnography (the “gold standard” of sleep diagnostics) and actigraphy [60,61], these are often only available to a limited extent in the clinical setting and particularly with PSG impose a potentially substantial burden and stress on patients and families (e.g., unfamiliar environment, need to stay overnight in clinic, etc., [61]). It has been emphasized that despite being correlated, actigraphy and sleep questionnaires/sleep logs may generate different estimates of the same sleep parameter, suggesting that these provide unique information contributing to the clinical understanding of patients’ sleep behavior and potential disorders [62]. Against this background, our study provides for the first time insights into a questionnaire that is even more specific than comparable instruments for CP patients and thus offers great potential for future use. In a next highly significant step, the questionnaire should be combined and compared with objective diagnostic measures, such as PSG or actigraphy, and with the impression of additional informants (e.g., professional caregivers [63]) to generate further evidence on the SNAKE and a comprehensive picture of the sleep behavior of this special population.

## 6. Conclusions

Our study showed that sleep peculiarities of children with CP seem to be primarily linked to their functional impairment. Nevertheless, these peculiarities do not necessarily represent sleep disorders, as the incorporation of a questionnaire developed specifically for this population showed. Follow-up studies should further elucidate these findings taking into account the limitations addressed.

## Figures and Tables

**Figure 1 children-08-00772-f001:**
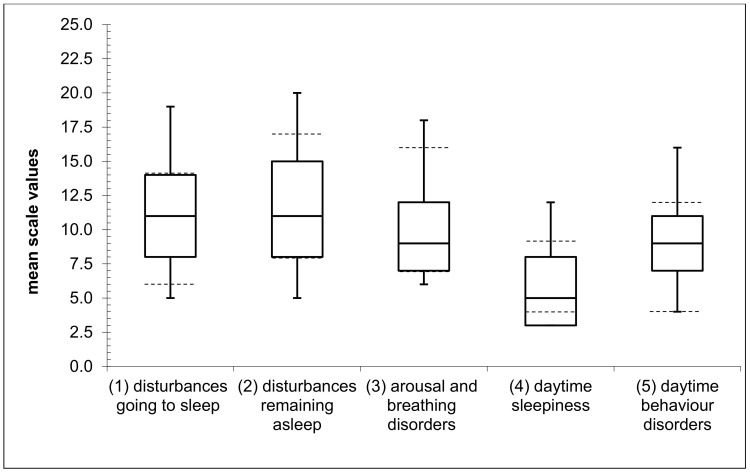
Box plot diagram for Sleep Questionnaire for Children with Severe Psychomotor Impairment (SNAKE) scales 1–5 (dashed lines indicate the upper and lower ends of the respective normal range).

**Figure 2 children-08-00772-f002:**
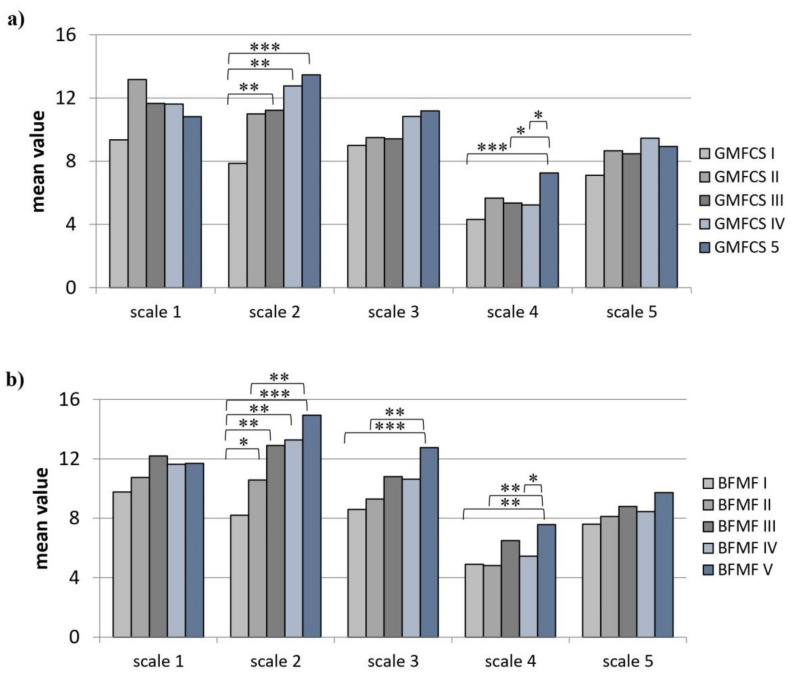
Differences between the five levels of Gross Motor Function Classification System (GMFCS) (**a**) and Bimanual Fine Motor Function (BFMF) (**b**) for the SNAKE scales 1–5. 1. disturbances going to sleep, 2. disturbances remaining asleep, 3. arousal and breathing disorders, 4. daytime sleepiness, 5. daytime behavior disorders. * *p* < 0.05, ** *p* < 0.01, *** *p* < 0.001.

**Table 1 children-08-00772-t001:** Patient characteristics (N = 100).

Characteristics	*n* (%) *
**Type of CP**	
BSCP	57 (57)
USCP	29 (29)
dyskinetic	14 (14)
*dystonic*	13
*choreoathethoid*	1
**cause of CP**	
periventricular leukomalacia	25 (25)
ischemic stroke	13 (13)
hypoxic brain injury	13 (13)
brain malformations	13 (13)
infections	10 (10)
unknown	10 (10)
intracranial bleeding	8 (8)
genetic factors	7 (7)
traumatic brain injury	1 (1)
**CP comorbidities**	
epilepsy	58 (58)
use of anticonvulsants	47 (81)
visual impairment	65 (65.7)
*severe*	15 (23.8)
cognitive impairment	56 (71.8)
hearing impairment	23 (23.2)
*severe*	6 (26.1)
**birth weight**	
>2500 g	51 (53.1)
1500–2500 g	18 (18.8)
1000–1499.99 g	16 (16.7)
<1000 g	11 (11.5)
**gestational age**	
term born (37–41 weeks)	51 (51.5)
extreme preterm (<32 weeks)	32 (32.3)
preterm (32–36 weeks)	15 (15.2)
postterm (>41 weeks)	1 (1)
**type of delivery**	
C-section, emergency	41 (41)
vaginal	34 (34)
C-section, planned	25 (25)
**GMFCS**	
I	29 (29)
II	6 (6)
III	21 (21)
IV	13 (13)
V	31 (31)
**BFMF**	
I	35 (35)
II	24 (24)
III	10 (10)
IV	11 (11)
V	20 (20)

* *n* may differ for individual characteristics; CP, cerebral palsy; BSCP, bilateral spastic cerebral palsy; USCP, unilateral spastic cerebral palsy; GMFCS, Gross Motor Functions Classification System; BFMF, Bimanual Fine Motor Function.

**Table 2 children-08-00772-t002:** Correlation of the SNAKE scale values with the CP subtype, different comorbidities, GMFCS, and BFMF.

SNAKE Scales	CP Characteristics							
	Subtype	Comorbidities						
		VI *severe*	HI *severe*	CI	Epilepsy	Anticonvulsants	GMFCS	BFMF
1	0.09	0.00	0.03	0.07	0.02	−0.11	0.13	0.20 *
		0.06	0.00					
2	0.02	0.21 *	0.17	0.32 *	0.27 *	0.07	0.49 **	0.57 **
		0.30 *	0.03					
3	−0.12	0.12	0.27 *	0.15	0.24 *	−0.05	0.24 *	0.41 **
		0.14	−0.00					
4	0.06	0.15	−0.05	0.16	0.22 *	0.04	0.39 *	0.34 **
		0.27 *	−0.13					
5	0.02	0.15	0.05	0.23 *	0.23 *	0.11	0.21 *	0.21 *
		0.15	−0.09					

1. disturbances going to sleep, 2. disturbances remaining asleep, 3. arousal and breathing disorders, 4. daytime sleepiness, 5. daytime behavior disorders. CP, cerebral palsy; VI, visual impairment; HI, hearing impairment; CI, cognitive impairment. * *p* < 0.05, ** *p* < 0.01.

## Data Availability

The datasets used and/or analyzed within the framework of this study are available from the corresponding author on reasonable request.

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
