# Peer review of "Assessment of Sleep-Related Problems in Children with Cerebral Palsy Using the SNAKE Sleep Questionnaire"

_children, 2021, doi:10.3390/children8090772_

Round 1

Reviewer 1 Report

General comments:

The authors investigated the sleep-related problems in children with cerebral palsy (CP) using the Sleep Questionnaire for Children with Severe Psychomotor Impairment (SNAKE). The theme of this study is important to understand the sleep condition in this population. However, there are serious concerns in the methods. These concerns make it difficult to interpret the results and draw the conclusion.

Major comments:

  1. Concerns in the study population

This study included from toddler to puberty. This age range is too wide. In addition, children with CP who were admitted to the Pediatric Neurology ward of the Children’s and Adolescents’ Hospital were included. However, there was no data about the reason for admission, and it was likely to be the fairly-biased study population.

  1. Concerns in the evaluation tools

The author stated that the utility of SNAKE for examining sleep in children with cerebral palsy was not well validated. However, there was no data about actual measured values about sleep. In addition, there was no data about other sleep questionnaire which was well validated in this population. Therefore, the reliability of data in this study seems to be low.

  1. Concerns in the patient characteristics

Cardiorespiratory abnormalities such as chronic heart failure, chronic respiratory failure, and obstructive sleep apnea syndrome affected sleep conditions in children with CP. In addition, musculoskeletal abnormalities including scoliosis and psychiatric disorders affected sleep conditions in children with CP. Tube feeding and urethral catheterization were also associated with sleep problems. However, there was no data about these abnormalities which were common in children with CP in this manuscript. Moreover, there was no information on drugs other than anticonvulsants.

  1. Sample size

The author stated that the overall sample size was sufficient in the limitation section. However, there was no data about desired sample size. Please calculate the desired sample size about each statistical method. I think it is insufficient for desired sample size because this study is based on not actual measured data but data from questionnaire.

Author Response

Assessment of sleep- related problems in children with cerebral palsy using the SNAKE sleep questionnaire

 children-1331781

 General comments:

The authors investigated the sleep-related problems in children with cerebral palsy (CP) using the Sleep Questionnaire for Children with Severe Psychomotor Impairment (SNAKE). The theme of this study is important to understand the sleep condition in this population. However, there are serious concerns in the methods. These concerns make it difficult to interpret the results and draw the conclusion.

 Thank you for your recognition of the pertinence of the subject matter of our study.

Major comments:

  1. Concerns in the study population

This study included from toddler to puberty. This age range is too wide. In addition, children with CP who were admitted to the Pediatric Neurology ward of the Children’s and Adolescents’ Hospital were included. However, there was no data about the reason for admission, and it was likely to be the fairly-biased study population.

Thank you very much for your helpful comments and the opportunity to revise our paper with regard to your suggestions. We supplemented the analysis of potential differences in participants' sleep behavior with a subgroup analysis according to their age:

  • 4, ll. 150-152: “A Kruskal- Wallis test followed by pairwise comparisons were used to determine potential differences in the sleep patterns among age groups which were defined as follows: toddler (age 2-3 years), child (age 4-12 years), adolescent (age 13-18 years).”
  • 6, ll. 222- 226: “On scale 4 (daytime sleepiness), a significant difference was identified in the sleep patterns of the participants depending on their age. Pairwise comparisons showed that toddlers achieved significantly higher SNAKE scale 4 values than the comparison groups children (χ2= 28.57, p< 0.01), and adolescents (χ2= 30.98, p< 0.01; detectable effect size: 0.36).
  • 10, ll. 387- 396: “Our study showed that toddlers showed a more conspicuous daytime sleepiness than children or adolescents. This contradicts existing literature, which identified more sleep problems in older children than in younger ones [45]. Since none of the other SNAKE scales showed corresponding effects, it is unlikely that this result can be explained by, for example, toddlers’ peculiar sleep regulation mechanisms. Possibly parents misinterpreted the questions about their child's sleep behavior during the daytime and considered "intentional" falling asleep like the nap, which is typical for this age group, as sudden (pathological) falling asleep during the daytime. Future projects should shed more light on such findings by collecting additional sleep-associated variables (e.g., does the child take naps, does the child take medication during the day).”

Furthermore, we have added information on the children’s reason for admission:

  • 2, ll. 87- 90: “Comprehensive screening was performed on all children with CP who were planned to be admitted to the Pediatric Neurology ward of the Children’s and Adolescents’ Hospital Datteln in Germany between the years 2015- 2017. Reasons for admission included symptom control (e.g., seizures), neuro- orthopedic assessments, cognitive testing or other issues such as targeted weight gain.”
  1. Concerns in the evaluation tools

The author stated that the utility of SNAKE for examining sleep in children with cerebral palsy was not well validated. However, there was no data about actual measured values about sleep. In addition, there was no data about other sleep questionnaire which was well validated in this population. Therefore, the reliability of data in this study seems to be low.

We have emphasized more clearly in the text that the SNAKE has good reliability and validity in children with psychomotor impairments and complex (neurological) conditions. At the same time, we have explicitly named other, established sleep questionnaires and better emphasized the important difference of these to the SNAKE:

  • 2, ll. 64- 72: “The “Sleep Questionnaire for Children with Severe Psychomotor Impairment (Schlaffragebogen für Kinder mit Neurologischen und Anderen Komplexen Erkrankungen, SNAKE)”, was specifically developed for children with psychomotor impairments and complex (neurological) conditions and was confirmed to show a good reliability and validity [26]. It comprises items that are disregarded in other frequently used questionnaires such as the “Sleep Disturbance Scale for Children” (SDSC; [21,27]), the “Children’s Sleep Habits Questionnaire” (CSHQ; [28-32]) or the “Sleep Behavior Questionnaire” (SBQ; [33,34]) that have been applied in different studies [35], but, however, were developed per se for healthy persons [26,36,37]. Although the potential utility of the SNAKE for examining sleep in children with CP has been pointed out [38], it has not yet been used in a systematic study with this particular population.

The importance of including more objective data collection instruments in future projects has been integrated in the Limitations section:

  • 11, ll. 446- 449: “Lastly, the results of our study are based on the judgment of the children's parents. Although these are "experts" for their child, it would be preferable in the future to validate their statements by including additional informants (e.g., professional caregivers [58]) or more objective instruments such as actigraphy or polysomnography (PSG).”
  1. Concerns in the patient characteristics

Cardiorespiratory abnormalities such as chronic heart failure, chronic respiratory failure, and obstructive sleep apnea syndrome affected sleep conditions in children with CP. In addition, musculoskeletal abnormalities including scoliosis and psychiatric disorders affected sleep conditions in children with CP. Tube feeding and urethral catheterization were also associated with sleep problems. However, there was no data about these abnormalities which were common in children with CP in this manuscript. Moreover, there was no information on drugs other than anticonvulsants.

 Thank you for pointing out this lack of potentially relevant information. We have included corresponding considerations in the discussion:

  • 11, ll. 439- 445: “The sleep behavior of children with CP may be influenced by more parameters that were not evaluated in detail within the scope of this study. For example, information on the (artificial) respiration of a child, on the child’s entire medication, on cardiorespiratory or musculoskeletal abnormalities could allow an even more precise interpretation of the study results. In future studies, background variables such as these that were shown to be potentially relevant for the sleep behavior of children with CP should be included to constantly generate more knowledge about such condition [6,56,57].”
  1. Sample size

The author stated that the overall sample size was sufficient in the limitation section. However, there was no data about desired sample size. Please calculate the desired sample size about each statistical method. I think it is insufficient for desired sample size because this study is based on not actual measured data but data from questionnaire.

Thank you for this comment. We did not perform an a priori sample size calculation and have mentioned this limitation in the text. To show the sufficiency of the sample size we also added sensitivity analyses to determine the detectable effect sizes for all tests:

  • 3, ll. 143- 146: “No a priori sample size calculation was conducted. The sample size was set to N= 100. Due to missing sample size calculation, sensitivity analyses were performed for each test via the software G*Power [46]. For this, a small to medium effect size and sufficient statistical power of 0.9 was assumed by default.”
  • 6, ll. 206- 209: “The BFMF additionally showed a significant association with hearing impairment (Fisher-Freeman-Halton exact test; detectable effect size: 0.4; after Bonferroni- Holm- correction all p< 0.05).”
  • 6, ll. 222- 226: “On scale 4 (daytime sleepiness), a significant difference was identified in the sleep patterns of the participants depending on their age. Pairwise comparisons showed that toddlers achieved significantly higher SNAKE scale 4 values than the comparison groups children (χ2= 28.57, p< 0.01), and adolescents (χ2= 30.98, p< 0.01; detectable effect size: 0.4).”
  • 8, ll. 279- 282: “Children with the least gross motor impairment achieved significantly lower scale values on scale 2 (disturbances remaining asleep) than those with more profound motor deficits (χ2= 26.79, p< 0.01; pairwise comparisons: GMFCS 1-3, GMFCS 1-4, GMFCS 1-5; all p< 0.05), as can be seen in Figure 2 (detectable effect size: 0.4).”
  • 7, ll. 261- 262: “None of the SNAKE scales showed an association with anticonvulsant medication intake (Table 2; detectable effect size: 0.3).”
  • 9, ll. 346- 348: “After Bonferroni correction, conducted Mann-Whitney-U tests (detectable effect size: 0.6) showed no differences between children experiencing severe visual impairment, reported cognitive impairment or epilepsy and those children who did not.”

Likewise, we have included a section on this aspect in the Discussion:

  • 11, ll. 428- 436: “Another limitation is that the study's sample size (N= 100) was not based on a priori calculations. Even though the sensitivity calculations showed that sufficient effect sizes and thus interpretable and meaningful results could be expected for all tests of interest, corresponding a priori calculations should be performed in future replicating studies. In this context, it is nevertheless important considering that especially in highly specialized settings such as the one focused on in this study, researchers are always faced with the challenges of small samples [59]. Researchers should therefore always weigh realistically between the desired sample sizes and effects and the actual sample that is presumably to be reached.”

Reviewer 2 Report

Dear Authors:

Thank you so much for your research. I think that it's a fantastic research about one of the CP comorbidity and it's a very important topic in CP's quality of life.

I write some suggestions:

  • Abstract à this type of information does not make much sense so I think that you can delete it: “Periventricular leukomalacia represented the most common cause for CP (n = 25), bilateral spastic 25 cerebral palsy (BSCP) the most common CP subtype (n = 57). The majority of children met a GMFCS 26 level 5 (n = 31)”.
  • Apart from the two classification systems, the non-use of MACS or VFCS is not understood when evaluating the visual system. Can you add this information? Is it possible?
  • Line 54, add difficulties in achieving learning due to sleep disturbance
  • Line 106 to 117, it is not necessary to specify all the levels …… ..or use table instead of text to make it clearer
  • Data analysis would go within the methodology
  • Does CPAP use persist in all children? It would be interesting to see if these children with CP have oxygenation difficulties.

Regards

Author Response

Assessment of sleep- related problems in children with cerebral palsy using the SNAKE sleep questionnaire

children-1331781

Dear Authors:

Thank you so much for your research. I think that it's a fantastic research about one of the CP comorbidity and it's a very important topic in CP's quality of life.

I write some suggestions:

Abstract à this type of information does not make much sense so I think that you can delete it: “Periventricular leukomalacia represented the most common cause for CP (n = 25), bilateral spastic 25 cerebral palsy (BSCP) the most common CP subtype (n = 57). The majority of children met a GMFCS 26 level 5 (n = 31)”.

Thank you very much for your helpful comments and the opportunity to revise our paper with regard to your suggestions. We deleted the corresponding passage in the abstract.

Apart from the two classification systems, the non-use of MACS or VFCS is not understood when evaluating the visual system. Can you add this information? Is it possible?

Thank you very much for this advice. We have integrated a corresponding rationale into the methodology to facilitate the reader's understanding:

  • 3., ll. 110- 114: “For the assessment of the children's cognitive, visual and hearing abilities, the definitions of the SCPE Data Collection Form for Cerebral Palsy were considered adequate for the study purpose, so that more complex assessment instruments such as the Manual Ability Classification System (MACS, [33]) did not need to be involved.”

The Visual Function Classification System (VFCS) was not yet published at the time of our data collection. We would prefer not to include this information in the text, as it can be confusing. However, we are very happy to note this instrument as a potentially useful tool for our upcoming projects.

Line 54, add difficulties in achieving learning due to sleep disturbance

We have added this information:

  • 2., ll. 54-57: “Sleep problems are not only having a major impact (e.g.; learning or behavioral difficulties) on the well-being of the affected child but are also frustrating for parents and siblings and may lead to a deterioration of their own sleep [6,11,17-19].”

Line 106 to 117, it is not necessary to specify all the levels …… ..or use table instead of text to make it clearer

Thank you for this comment; we deleted the detailed information on the GMFCS and BFMF.

Data analysis would go within the methodology

We have assigned the data analysis to the methodology and adjusted the numbering accordingly.

Does CPAP use persist in all children? It would be interesting to see if these children with CP have oxygenation difficulties.

Unfortunately, the data structure does not allow us to evaluate such information. Parents were solely questioned whether their child was ventilated with a respirator or CPAP in the NICU (if applicable).

Nonetheless, we included this interesting aspect in the limitations section as an idea for future research efforts:

  • 11, ll. 439-445: “The sleep behavior of children with CP may be influenced by more parameters that were not evaluated in detail within the scope of this study. For example, information on the (artificial) respiration of a child, on the child’s entire medication, on cardiorespiratory or musculoskeletal abnormalities could allow an even more precise interpretation of the SNAKE results. In future studies, background variables such as these that were shown to be potentially relevant for the sleep behavior of children with CP should be included to constantly generate more knowledge about such condition [6,56,57].

Round 2

Reviewer 1 Report

Thank you for your prompt and polite reply to my review comments. I think you responded appropriately to concerns in the study population, SNAKE, and sample size. However, serious concerns about the lack of actual measured values about sleep and the lack of information about patient characteristics have not been improved. 

Author Response

Thank you for your prompt and polite reply to my review comments. I think you responded appropriately to concerns in the study population, SNAKE, and sample size. However, serious concerns about the lack of actual measured values about sleep and the lack of information about patient characteristics have not been improved.

Thank you for your quick feedback and your comments. We absolutely agree with you that actual, objective sleep data is extremely important and useful in research with CP patients. However, equally, subjective instruments such as the SNAKE provide meaningful information on the sleep behavior of (CP) patients. With our study, we aim to present researchers and practitioners a valid, easy-to-use instrument specifically designed for patients with neurological and other complex disorders and psychomotor impairments, which provides significant advantages over sleep questionnaires initially developed for healthy patients. We have listed these aspects more thoroughly in the discussion:

  • 10, ll. 388- 406: “Lastly, the sleep- related results of our study are based on the judgment of the children's parents. Given that these are experts on their child and that the application of sleep questionnaires - albeit not specifically developed for children with neurological disorders and psychomotor impairments - is a frequently chosen method in children and patients with CP [54], this should not per se be considered a weakness of the study. Even if there are more objective procedures such as polysomnography (the “gold standard” of sleep diagnostics) and actigraphy [60,61]), these are often only available to a limited extent in the clinical setting and particularly with PSG impose a potentially substantial burden and stress on patients and families (e.g., unfamiliar environment, need to stay overnight in clinic, etc., [61]). It has been emphasized that despite being correlated, actigraphy and sleep questionnaires/sleep logs may generate different estimates of the same sleep parameter, suggesting that these provide unique information contributing to the clinical understanding of patients’ sleep behavior and potential disorders [62]. Against this background, our study provides for the first-time insights into a questionnaire that is even more specific than comparable instruments for CP patients and thus offers great potential for future use. In a next highly significant step, the questionnaire should be combined and compared with objective diagnostic measures such as PSG or actigraphy and with the impression of additional informants (e.g., professional caregivers [63]) to generate further evidence on the SNAKE and a comprehensive picture of the sleep behavior of this special population.”

In the discussion, we emphasized more strongly that although the characteristics and variables we focused on were sufficient to answer our research questions, it is essential to consider additional patient characteristics and variables in the future and to continuously generate new insights into sleep in CP patients.

  • 10, ll. 381- 394: “The sleep behavior of children with CP may be influenced by more parameters and patient characteristics that were not evaluated in detail within the scope of this study. For example, information on the (artificial) respiration of a child, on the child’s entire medication, on cardiorespiratory or musculoskeletal abnormalities were not focused in detail as we were primarily interested in variables that are recorded as part of the SCPE Data Collection Form and by the SNAKE. For the purpose of the study, the selected variables are to be considered satisfactory, as the main objective of the study and our corresponding questions could be answered. Nonetheless, CP is a highly complex disorder [3], so the collection of additional patient characteristics and variables is imperative in future studies to continuously better understand sleep behavior and potential sleep problems in this population [6,57,58].”

Round 3

Reviewer 1 Report

The authors respond politely to my review comments